# Association between Problematic Internet Use and Sleep Disturbance among Adolescents: The Role of the Child’s Sex

**DOI:** 10.3390/ijerph15122682

**Published:** 2018-11-28

**Authors:** Jiewen Yang, Yangfeng Guo, Xueying Du, Yi Jiang, Wanxin Wang, Di Xiao, Tian Wang, Ciyong Lu, Lan Guo

**Affiliations:** 1Health Promotion Centre for Primary and Secondary Schools of Guangzhou Municipality, Guangzhou, 510080, China; zhengyi6888@163.com (J.Y.); guoyangfeng666@yeah.net (Y.G.); duxueying@yeah.net (X.D.); jiangyi388@yeah.net (Y.J.); 2Department of Medical statistics and Epidemiology, School of Public Health, Sun Yat-sen University, Guangzhou 510080, China; wgg0808@163.com (W.W.); xiaodi@mail2.sysu.edu.cn (D.X.); wangt97@mail2.sysu.edu.cn (T.W.)

**Keywords:** problematic Internet use, sleep disturbance, sex difference, adolescents

## Abstract

Use of the Internet has become an integral part of daily life. Adolescents are especially at a higher risk of developing problematic Internet use (PIU). Although one of the most well-known comorbid conditions of PIU is sleep disturbance, little is known about the sex disparity in this association. This school-based survey in students of grades 7–9 was conducted to estimate the prevalence of PIU and sleep disturbance among Chinese adolescents, to test the association between PIU and sleep disturbance, and to investigate the role of the child’s sex in this association. A two-stage stratified cluster sampling method was used to recruit participants, and two-level logistic regression models were fitted. The mean Internet addiction test score was 37.2 (SD: 13.2), and 15.5% (736) met the criteria for PIU. After adjusting for control variables, problematic Internet users were at a higher risk of sleep disturbance (adjusted odds ratio = 2.41, 95% confidence interval (CI) = 2.07–3.19). Sex-stratified analyses also demonstrated that association was greater in girls than boys. In this respect, paying more attention to the sleep patterns of adolescents who report excessive Internet use is recommended, and this early identification may be of practical importance for schools, parents, and adolescents themselves.

## 1. Introduction

Internet use has increased rapidly all over the world, and the Internet has become an integral part of daily life (e.g., in communication, education, and entertainment) [1]. However, overuse of Internet can cause damage to psychological function, many maladaptive problems, and even problematic Internet use (PIU, also termed as Internet addiction) [2,3]. Adolescence is the transition period from puberty and adulthood; events during this period can have influences on an individual’s development and can determine their behavior and attitudes in later life [4]. Indeed, most adolescents are exceptionally vulnerable and receptive, choosing Internet use as a release to cope with unpleasant feelings and emotional crises, instead of offline interaction with peers and parents [5]. Given their vulnerable age, adolescents are especially at a higher risk of developing PIU. Although the reported prevalence of PIU varies in different countries, adolescent PIU has become a global public health issue [6], and China is no exception.

One of the most well-known comorbid conditions of PIU is sleep disturbance [7]. Sleep has an important role in adolescent lives, and is necessary for physical and mental health across adolescence [8]. However, evidence suggests that sleep problems tend to increase during adolescence, and poor sleep quality and sleep disturbance among adolescents has been a major public health concern [9]. Sleep deprivation is related to low self-control that has been found to be associated with a range of behaviors including PIU among adolescents [10,11,12]. Prior studies reported that adolescents who spent more time using the Internet had less sleeping time, which may cause irregular sleep patterns [13]. PIU among adolescents was also found to be associated with insomnia and the disturbance of sleep [14,15]. “Time displacement theory” has been used to explain that Internet use, in general, reduces the time for other activities (including sleep) [16]. Considering the brain undergoes rapid and extensive development during adolescence, it is more vulnerable to the negative consequences of PIU and sleep disturbance that can cause neurological and behavioral changes.

Although some studies reported the significant associations between PIU and sleep disturbance [17], little is known about the sex disparity in this association among Chinese adolescents. Several lines of evidence showed that slightly more males than females used the Internet regularly [18], more female than male adolescents used the Internet for school/work significantly [19], and males are more likely to be addicted to the Internet compared to females [20]. Additionally, sex differences in sleep become apparent after the onset of puberty, and female sex hormones are reported to have more significant effects on the sleep–wake cycle when compared to male sex hormones [21]. Therefore, we hypothesized that the child’s sex may play a role in the association between PIU and sleep disturbance. There has been a rapid socio-economic change in China during the past three decades, and that may have also affected the sleep patterns and PIU among Chinese adolescents. This study was conducted to estimate the prevalence of PIU and sleep disturbance among Chinese adolescents, to test the association between PIU and sleep disturbance, and to investigate the role of the child’s sex in this association.

## 2. Materials and Methods

### 2.1. Study Design and Participants

This was a school-based survey in students of grades 7–9 (i.e., middle school) attending public or private schools in Guangzhou. In 2014, a two-stage stratified cluster sampling method was used to recruit participants. In stage 1, there were a total of 11 districts in Guangzhou, and one middle school (or primary sampling units) was randomly selected from each district. In stage 2, two classes (or secondary sampling units) were randomly selected from each grade within the selected schools. All available students in the selected classes were invited to participate in our study. Of the 4930 school students who were invited to participate, 4750 students’ questionnaires were completed and qualified for the survey (a response rate of 96.3%). Informed consent letters were obtained from one of the students’ parents or other responsible adult after the nature of the study was explained, and the conduct of the study followed the tenets of the Declaration of Helsinki. The study was approved by the Sun Yat-sen University, School of Public Health Institutional Review Board (ethics code: L2014076).

### 2.2. Questionnaire

Researcher assistants visited the school to distribute questionnaires, and each participating student completed the questionnaires in the classrooms without the presence of teachers. Information on sociodemographic factors were collected, including the child’s sex (1 = male, 2 = female), age, ethnicity (1 = Han, 2 = other ethnic group), household socioeconomic status (HSS, as assessed by the students’ self-rating of their family’s economic status with responses were coded as 1 = above average, 2 = average, 3 = below average), and academic pressure (as measured by asking students about their perceptions of his or her own academic pressure, with responses coded as 1 = above average, 2 = average, 3 = below average). Student’s outdoor activity was measured by asking how many hours the child spent in outdoor activities per week.

Body weight and height were measured by standard anthropometric methods. According to the standardized growth charts from the Working Group on Obesity for Children (WGOC) in China, body mass index (BMI) z scores (standard deviation: SD) for each participant were calculated to represent the deviation compared with an average child of the same sex and age [22]. Students with a BMI z score less than the 85th percentile were considered “normal”, students with a BMI z score between the 85th and 95th percentile were considered “overweight”, and those with a BMI z score above the 95th percentile were thought to be “obese” [23]. In this study, weight status was coded as 1 = normal, 2 = overweight or obese.

Problematic Internet use (PIU) was assessed using the Chinese version of the Young’s Internet Addiction Test (IAT); the IAT consists of 20 items rated in a five-point Likert scale (from 1 = not at all to 5 = always) [24]. The Chinese version has been validated in Chinese adolescents with satisfactory psychometric properties [25,26], and the Cronbach’s alpha for IAT was 0.93 in the present study. Total score of the IAT ranges from 20 to 100 and represents an individual’s tendency to PIU. The higher score suggests the greater level of PIU. In this study, we not only described the total IAT scores but also used the validated cut-off value of 50 [27].

Sleep quality and disturbances over a 1-month time interval was measured using the Chinese version of Pittsburgh Sleep Quality Index (PSQI) which has been validated in Chinese samples, and the Cronbach’s alpha for PSQI was 0.86 in the present study. The sum of the scores for these seven components yields one global score with a range of 0–21 points in which higher scores indicate worse sleep quality [28,29]. A PSQI global score of above 7 points indicates poor sleep quality collectively known as sleep disturbance.

### 2.3. Statistical Analysis

All statistical analyses were conducted using SAS 9.2 (SAS Institute, Inc., Cary, NC, USA). To assess any differential relationships across the child’s sex, sex-combined and sex-specific analyses were performed. First, descriptive analyses (also conducted separately for boys and girls) were used to describe the sample characteristics, and t tests for continuous variables and Chi-square tests for categorical variables were performed. Second, considering this study utilized a two-stage sampling design in which students were grouped into classes, two-level logistic regression models were fitted in which classes were treated as clusters. The variables that were significant at 0.10 level in the univariate analyses or widely reported in the literature were simultaneously incorporated in the two-level multivariate logistic regression models to test the independent associations of PIU with sleep disturbance. Additionally, to test the interaction effects of sex and PIU on sleep disturbance, the variables that were significant at 0.10 level in the univariate analyses or widely reported in the literature, the child’s sex, PIU, and the interaction item between sex and PIU were simultaneously entered into a two-level multivariate logistic regression model [30]. In these regression models, the outcome variables were the presence of sleep disturbance. Regarding logistic regression analyses, observations with missing data were eliminated (less than 6.0%). Statistical significance was evaluated at the <0.05 level using two-sided tests.

## 3. Results

The sample characteristics are shown in Table 1. Of the total sample, 49.2% (2335) were boys and 50.8% (2415) were girls, yielding a male-to-female ratio of 1:1.03. The mean age of the students was 16.0 (SD: 1.5). The proportion of students who reported above average HHS was 33.3%, the proportion of students who admitted above average academic pressure was 51.0%, and the proportion of overweight or obesity in students was 15.2%. The majority of students spent less than 2 h a day doing outdoor activity (69.8%). The mean IAT score was 37.2 (SD: 13.2), and 15.5% (736) met the criteria for PIU. The mean PSQI score was 6.8 (SD: 3.5), and 39.1% (1863) met the criteria for sleep disturbance. The distributions of HSS, academic pressure, weight status, daily hours of outdoor activity, IAT scores, PIU status, PSQI scores, and sleep disturbance status between boys and girls were statistically significant (*p* < 0.05).

As shown in Table 2, without adjusting for other variables, PIU, age, ethnicity, HSS, academic pressure were significantly associated with sleep disturbance in all students. Additionally, sex-stratified analyses also demonstrated that similar associations can be found in boys and girls separately.

As shown in Table 3, after incorporating sex, PIU, the interaction item between sex and PIU, ethnicity, HSS, academic pressure in the two-level multivariate logistic regression model, the results showed that the interaction item between sex and PIU was significantly associated with sleep disturbance (*p* < 0.05). Moreover, as shown in Figure 1, after adjusting for age, sex, ethnicity, HSS, academic pressure, problematic Internet users were at a higher risk of sleep disturbance (adjusted odds ratio (aOR) = 2.41, 95% confidence interval (CI) = 2.07–3.19). Additionally, the magnitude of aORs for the association between PIU and sleep disturbance (aOR = 3.19, 95% CI = 2.34–4.34) was greater in girls than boys (aOR = 2.07, 95% CI = 1.60–2.68).

## 4. Discussion and Implication

This study found that 15.5% of adolescents in Guangzhou met the criteria for PIU, and this prevalence is higher than that described in our prior study conducted in 2011 showing that 12.2% were identified as problematic Internet users [26]. Additionally, a recent study showed that the overall of PIU among adolescents in Guangzhou was 26.0% [31]. These results may indicate that PIU has been a growing problem among Chinese adolescents. Our study also found that 39.1% of adolescents met the criteria for sleep disturbance (defined as PSQI > 7). This finding is aligned with our prior study suggesting that sleep disturbance was not rare among Chinese adolescents, with a prevalence of 39.6% [32]. Similarly, a prior Brazilian study also reported that 38.9% of the students had sleep disturbance as measured by the PSQI [33]. Moreover, the present study showed that PIU was associated with an elevated risk of sleep disturbance, and there existed significant differences between the magnitude of the association in girls and in boys.

Sleep disturbance can be harmful, and influences hormonal function resulting in psychological and physiological impairment [34]. In line with previous evidence [11,12,32,35,36], the present study first showed that without adjusting for other variables, PIU was positively associated with sleep disturbance. Older adolescents were at a higher risk of sleep disturbance. Adolescents who reported average or below average HSS were more likely to report having sleep disturbance than those with above average HSS. Adolescents with average or below average academic pressure were less likely to be involved in sleep disturbance than the group with above average academic pressure. These findings may provide a basis for identifying adolescents at high risk for sleep disturbance, special attention should be paid to those students who present with the adverse characteristics mentioned above.

The present study first found that the interaction item between the child’s sex and PIU was significantly associated with sleep disturbance after adjusting for ethnicity, HSS, and academic pressure. Additionally, after adjusting for age, sex, ethnicity, HSS, and academic pressure, our final multivariate two-level logistic regression results demonstrated that adolescents who met the criteria for PIU were at an elevated risk of sleep disturbance. Similarly, Ekinci et al. reported that Turkish adolescents with a higher IAT score were more likely to get to bed later in night and need more time to fall asleep [15]; Lemola et al. found that electronic media use was negatively associated with sleep duration and positively with sleep difficulties among high school students in northwestern Switzerland [37]; Tan et al. also showed that PIU was strongly associated with sleep disturbance among high school students in southern China [38]; Kim et al. found that less sleep was significantly related to a long Internet use time for leisure among middle school students in Korea [39]. One possible explanation is related to the “time displacement theory” which postulates that PIU may displace sleep. In particular, using Internet for unstructured leisure activity with no fixed starting and stopping point may increase the risk of expanding more time and thus displacing other possible activities and sleep. Another possible explanation is that light emission from the devices for Internet use might interfere with sleep, and there is experimental evidence showing that LED back light screens emit an elevated amount of light in the short wave length that suppresses melatonin secretion in the evening and reduces objective and subjective signs of sleepiness [40]. Moreover, the results of the separate analyses by sex demonstrated that the adjusted associations between PIU and sleep disturbance appeared slightly stronger in girls than boys. These results might be related to that sleep and mood may share common genetic/molecular regulatory networks [41], and girls are more inclined to have mood disorders (e.g., depression and anxiety), leaving them more vulnerable to sleep problems [42]. Additionally, girls tend to use the Internet for school/work, but boys are more inclined to use the Internet for entertainment; the purpose of Internet use may have influence on the level of PIU [19]. According to the report of National Sleep Foundation, females tend to need more sleep than males [43], and girls are more likely to have more sleep problems than boys [44]. In the present study, the proportion of girls with sleep disturbance is also higher than that in boys. Moreover, a potential biological plausibility is that sex hormones and natural hormonal cycles may be responsible for the sex differences, and fluctuations of female sex hormones may have more significant effects on the sleep-wake cycle [21]. Based on the study results, several recommendations are proposed: (1) educational campaigns are need to improve the awareness of families, schools, and individuals on the negative consequences of PIU and sleep disturbance; (2) parents are encouraged to take strategies (e.g., using management software) to monitor and control their children’s Internet use; (3) schools are recommended to develop a suitable school schedule, considering most high schools start earlier than 07:00 h in China; (4) parents and schools need to support adolescents develop healthy sleep habits and Internet use patterns; and (5) a national long-term surveillance system (e.g., the Youth Risk Behavior Surveillance System (YRBSS)) is necessary to supervise the health-related behaviors (e.g., PIU and sleep disturbance) among adolescents in China.

This is a preliminary and exploratory study to test the association between PIU and sleep disturbance and to investigate the role of the child’s sex in this association. There are some methodological limitations that should be considered. First, due to the cross-sectional nature of the study, the observed associations should not be construed as causal. Second, as this is a school-based study, our study sample included only school students and did not recruit adolescents who had dropped out of school or were not present in school on the day the survey was administered. However, PIU or sleep disturbance may be more common among those who were absent. Third, mood disorder were not taken into consideration in the present study. Despite these limitations, our study sample was school-based, with a high participation rate and statistical randomization in recruitment. Additionally, given our study utilized a complex sampling design (a two-stage stratified cluster sampling method), two-level logistic regression analyses were performed to disentangle the within-cluster effects [45]. Most similar studies using multi-stage sampling method did not adopt this analysis method.

## 5. Conclusions

In conclusion, our study found a positive association between PIU and sleep disturbance, and the interaction item between the child’s sex and PIU was significantly associated with sleep disturbance. Moreover, the magnitude of the association between PIU and sleep disturbance is slightly stronger in girls than boys. This finding is an important addition to existing literature. Based on the findings of the present study, effective prevention and intervention strategies are recommended to be established. We should pay more attention to the sleep patterns of adolescents (especially girls) who report excessive Internet use, and this early identification may be of practical importance for schools, parents, and adolescents themselves.

## Figures and Tables

**Figure 1 ijerph-15-02682-f001:**
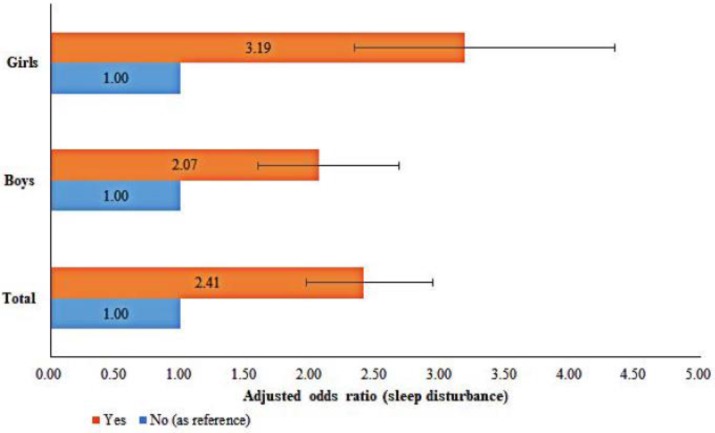
Adjusted association between problematic Internet use (PIU) and sleep disturbance: stratified by sex.

**Table 1 ijerph-15-02682-t001:** Sample characteristics stratified by sex.

Variable	Total (%)	Boys (%)	Girls (%)	*p*-Value
Total	4750 (100)	2335 (49.2)	2415 (50.8)	0.151
Age, mean (SD)	16.0 (1.5)	15.99 (1.5)	16.1 (1.5)	
Ethnicity				0.492
Han	4361 (91.8)	2137 (91.5)	2224 (92.1)	
Other ethnic groups	389 (8.2)	198 (8.5)	191 (7.9)	
HSS				<0.001
Above average	1583 (33.3)	764 (32.7)	819 (33.9)	
Average	2802 (59.0)	1340 (57.4)	1462 (60.5)	
Below average	306 (6.4)	205 (8.8)	101 (4.2)	
Missing data	59 (1.2)	26 (1.1)	33 (1.4)	
Academic pressure				<0.001
Above average	2424 (51.0)	1225 (52.5)	1199(49.6)	
Average	2073 (43.6)	955 (40.9)	1118 (46.3)	
Below average	229 (4.8)	141 (6.0)	88 (3.6)	
Missing data	24 (0.5)	14 (0.6)	10 (0.4)	
Weight status				<0.001
Normal	4027 (84.8)	1850 (79.2)	2177 (90.1)	
Overweight or obese	723 (15.2)	485 (20.8)	238 (9.9)	
Daily hours of outdoor activity				<0.001
Less than 2 h	3315 (69.8)	1463 (62.7)	1852 (76.7)	
2–3 h	731 (15.4)	428 (18.3)	303 (12.5)	
More than 3 h	536 (11.3)	347 (14.9)	189 (7.8)	
Missing data	168 (3.5)	97 (4.2)	71 (2.9)	
IAT scores, mean (SD)	37.2 (13.2)	38.5 (13.9)	35.9 (12.4)	<0.001
Problem Internet use				<0.001
No	4014 (84.5)	1906 (81.6)	2108 (87.3)	
Yes	736 (15.5)	429 (18.4)	307 (12.7)	
Total PSQI scores, mean (SD)	6.8 (3.5)	6.6 (3.7)	7.0 (3.2)	
Sleep disturbance				<0.001
No	2892 (60.9)	1494 (64.0)	1398 (57.9)	
Yes	1858 (39.1)	841 (36.0)	1017 (42.1)	

Abbreviations: HSS, household socioeconomic status; SD, standard deviation; IAT, Young’s Internet Addiction Test. Chi-squared tests were used for categorical variables, and t tests were used for age data, IAT scores, and total Pittsburgh Sleep Quality Index (PSQI) scores.

**Table 2 ijerph-15-02682-t002:** Unadjusted odds ratios and 95% confidence interval of sleep disturbance among adolescents: two-level logistic regression analyses.

Variable	Sleep Disturbance					
	Total, OR (95% CI)	*p*-Value	Boys, OR (95% CI)	*p*-Value	Girls, OR (95% CI)	*p*-Value
PIU (Ref. = No)						
Yes	2.57 (2.13–3.09)	<0.001	2.36 (1.85–3.02)	<0.001	3.15 (2.34–4.24)	<0.001
Age (1-year increase)	1.42 (1.35–1.48)	<0.001	1.48 (1.39–1.59)	<0.001	1.36 (1.28–1.44)	<0.001
Ethnicity (Ref. = Other ethnic groups)	1.29 (1.02–1.63)	0.038	1.30 (0.92–1.83)	0.139	1.28 (0.92–1.77)	0.146
HSS (Ref. = Above average)						
Average	1.63 (1.42–1.87)	<0.001	1.81 (1.48–2.21)	<0.001	1.48 (0.94–2.34)	0.091
Below average	2.35 (1.78–3.12)	<0.001	3.41 (2.37–4.91)	<0.001	1.48 (1.23–1.79)	<0.001
Academic pressure (Ref. = Above average)						
Average	0.52 (0.46–0.59)	<0.001	0.47 (0.38–0.57)	<0.001	0.56 (0.47–0.67)	<0.001
Below average	0.56 (0.41–0.76)	<0.001	0.59 (0.39–0.87)	0.008	0.55 (0.34–0.89)	0.015
Weight status (Ref. = Normal)						
Overweight or Obese	0.86 (0.72–1.02)	0.081	0.93 (0.74–1.16)	0.504	0.86 (0.64–1.15)	0.294
Daily hours of outdoor activity (Ref. = Less than 2 h)						
2–3 h	0.95 (0.79–1.13)	0.545	0.98 (0.78–1.25)	0.897	0.97 (0.75–1.27)	0.826
More than 3 h	0.91 (0.74–1.12)	0.380	0.87 (0.67–1.14)	0.309	1.13 (0.81–1.57)	0.484

Abbreviations: HSS, household socioeconomic status; PIU, problematic Internet use; Ref., reference; OR, odds ratio; 95% CI, 95% confidence interval.

**Table 3 ijerph-15-02682-t003:** Associations of the interaction item with sleep disturbance: a two-level multivariate logistic regression model *.

Variable	Sleep Disturbance	
	OR (95% CI)	*p*-Value
Sex * PIU (Interaction item)	1.48 (1.12–2.20)	0.031
PIU (Ref. = No)		
Yes	2.14 (1.67–2.75)	<0.001
Sex (Ref. = Boys)	1.32 (1.14–1.52)	<0.001

Abbreviations: PIU, problematic Internet use; Ref., reference; OR, odds ratio; 95% CI, 95% confidence interval. *: The child’s sex, PIU, the interaction item between sex and PIU, ethnicity, HSS, and academic pressure in the two-level multivariate logistic regression model.

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
