# Peer review of "Association between Problematic Internet Use and Sleep Disturbance among Adolescents: The Role of the Child’s Sex"

_ijerph, 2018, doi:10.3390/ijerph15122682_

Round 1

Reviewer 1 Report

The current version of the paper is suitable for the publicaion. But I think that the Authors should revise some statements in the results. In particular, the Authors mentioned their previous papers to comment the current results. I think that Authors should discuss the results by considering the others studies in this field.

Author Response

Response to Reviewer 1 Comments

Point 1: The current version of the paper is suitable for the publication. But I think that the Authors should revise some statements in the results. In particular, the Authors mentioned their previous papers to comment the current results. I think that Authors should discuss the results by considering the others studies in this field.

Response 1: Thank you for your kind suggestion. We have added other references to the revised manuscript (please see page 6, lines 175-176 & lines 180-181).

Reviewer 2 Report

This study explores the cross-sectional associations between problematic internet use (PIU) and sleep disturbance - a highly relevant field of study - in a large population of Chinese adolescents. The study has an exceptionally high response rate. The study also has the ambition to examine the role of sex for the association between PIU and sleep disturbance. The manuscript is well written and easy to follow (although some minor edits of the text are suggested, including a change of presentation in Table 1). My main concern is that if one of the major research questions deals with possible differences between the sexes, it is not enough to just stratify for sex and draw conclusions based on the size of the ORs. As can be seen in Figure 1 the confidence intervals of the girls’ and boys’ ORs overlap. The hypothesis should be statistically tested to see if the ORs for girls and boys differ (e.g., described by De Maris 2004). This can probably be pretty easily done, and the conclusions adapted thereafter. 

Introduction

General: The discussion is nice and swift, and easy to follow. But, has really no one looked at gender or sex differences in associations between PIU and sleep disturbance previously? I would think that stratifying for sex/gender was common.

P1 line 41. I suggest that when claiming that sleep disturbance is a well-known comorbid condition of PIU, a few references could be added.

P2 Line 49. Please add a reference to “Time displacement theory” and/or describe the theory.

Methods and materials

P2 Line 72. Check the use of singular/plural in the sentence.

P2 Questionnaire. How was information about family economic status and academic pressure collected? Did the adolescents respond to a question about this? Or was this information collected from the parents or registers.

P2 Questionnaire. In Results, weight status is reported. That weight status was collected should be mentioned in the Methods section. Was it only weight or was BMI measured? Self-report?

Results

P3:line 118: Please define HHS

P3 line 133-134: The aOR has a higher value for girls than for boys. However, the confidence interval is also wider and it overlaps with the boys’ confidence interval. Since the role of gender is part of the hypothesis/research question, I think it is insufficient to only stratify for sex (which all or most studies should do) and draw conclusions from the size of the ORs. It should be tested statistically if the girls' OR differs from the boys', in order to draw the conclusion that the association is actually stronger for the girls.  

Table 1: The percentages are presented differently in the columns. In the “Total” column, column percentages are used, while in the “Boys” and “Girls” columns, row percentages are used. The latter is confusing, and makes it difficult to compare the groups visually. I strongly suggest that column percentages are used overall.

Discussion

P5 General: Please summarize the main results (including the role of sex) early in the Discussion. Now, the results of one of the main research question (sex) is not presented until after some time (lines 180-181). However, here the correct terminology is used (“appeared stronger”) considering the lack of statistical testing of the potential difference of the ORs.

P6, lines 181--. In relation to the gender differences, some biological mechanisms are discussed. However, there is no discussion about potential differences in the content of internet use or behavioral aspects of internet use. Do the girls and boys use the internet for different purposes or in different manner (e.g. long bouts vs many short uses, or daytime/nighttime etc)? I suggest that the discussion about potential mechanisms for gender differences is slightly broadened.

General: The proportion of sleep disturbance (39%) in the study population is alarmingly high! Are sleep disturbances such a common problem for Chinese adolescents? Or is the cutoff at 7 for the PSQI perhaps not correct for a population of adolescents? However, I can see that if school starts earlier than 7:00 a.m., sleep deficiency is easily developed (especially since adolescents’ tend to have a circadium rhythm that favors late hours rather than early).

P6 line 186-- The authors propose several recommendations, based on the study results. Most of them are highly relevant. However, I do not see how number 5 is based on the study - (although relevant) it seems to relate more to general sleep behavior recommendations.  

Conclusions

See my previous notes about drawing conclusions about differences in ORs without statistically testing the difference. Either test the difference or tone down the conclusion.

Author Response

Response to Reviewer 2 Points

This study explores the cross-sectional associations between problematic internet use (PIU) and sleep disturbance - a highly relevant field of study - in a large population of Chinese adolescents. The study has an exceptionally high response rate. The study also has the ambition to examine the role of sex for the association between PIU and sleep disturbance. The manuscript is well written and easy to follow (although some minor edits of the text are suggested, including a change of presentation in Table 1). My main concern is that if one of the major research questions deals with possible differences between the sexes, it is not enough to just stratify for sex and draw conclusions based on the size of the ORs. As can be seen in Figure 1 the confidence intervals of the girls’ and boys’ ORs overlap. The hypothesis should be statistically tested to see if the ORs for girls and boys differ (e.g., described by De Maris 2004). This can probably be pretty easily done, and the conclusions adapted thereafter.

Introduction

Point 1: General: The discussion is nice and swift, and easy to follow. But, has really no one looked at gender or sex differences in associations between PIU and sleep disturbance previously? I would think that stratifying for sex/gender was common.

Response 1: Thank you for carefully and patiently reviewing our manuscript, and your insights were a tremendous help to us during this revision. According to your suggestion, this sentence has been changed to “although some studies reported the significant associations between PIU and sleep disturbance, little is known about the sex disparity in this association among Chinese adolescents” (please see page 2, line 55).

Point 2: P1 line 41. I suggest that when claiming that sleep disturbance is a well-known comorbid condition of PIU, a few references could be added.

Response 2: According to your suggestion, we have cited a reference (please see page 1, line 41).

Point 3: P2 Line 49. Please add a reference to “Time displacement theory” and/or describe the theory.

Response 3: Thank you for your kind suggestion. We have added a reference (please see page 2, line 50).

Methods and materials

Point 4: P2 Line 72. Check the use of singular/plural in the sentence.

Response 4: Thank you for noticing this. We apologize for this editing errors in our original manuscript. According to your suggestion, we have revised this sentence (please see page 2, line 72).

Point 5: P2 Questionnaire. How was information about family economic status and academic pressure collected? Did the adolescents respond to a question about this? Or was this information collected from the parents or registers.

Response 5: Thank you for your questions. In the original manuscript, we have reported that each participating student completed the questionnaires in the classrooms without the presence of teachers (please see page 2, lines 82-83). According to your suggestion, we have added the information about family economic and academic pressure to the revised manuscript (please see page 2, lines 85-89). Household socioeconomic status (HSS) was assessed by the students’ self-rating of their family’s economic status (responses were coded as 1=above average, 2=average, 3=below average), and academic pressure was measured by asking students their perception about his or her own academic pressure (responses were coded as 1=above average, 2=average, 3=below average).

Point 6: P2 Questionnaire. In Results, weight status is reported. That weight status was collected should be mentioned in the Methods section. Was it only weight or was BMI measured? Self-report?

Response 6: Thank you for your questions. In this study, body weight and height were measured by standard anthropometric methods. According to the standardized growth charts from the Working Group on Obesity for Children (WGOC) in China, body mass index (BMI) z scores (standard deviation: SD) for each participant were calculated to represent the deviation compared with an average child of the same sex and age [1]. Students with a BMI z score less than the 85th percentile were considered “normal”, students with a BMI z score between the 85th and 95th percentile were considered “overweight”, and those with a BMI z score above the 95th percentile were thought “obese” [2]. In this study, weight status was coded as 1=normal, 2=overweight or obese. These information have been added to the revised manuscript (please see page 3, lines 92-99).

Results

Point 7: P3: line 118: Please define HHS

Response 7: Thank you for your kind suggestion. We have revised this sentence to make it more clear (please see page 2, line 85).

Point 8: P3 line 133-134: The aOR has a higher value for girls than for boys. However, the confidence interval is also wider and it overlaps with the boys’ confidence interval. Since the role of gender is part of the hypothesis/research question, I think it is insufficient to only stratify for sex (which all or most studies should do) and draw conclusions from the size of the ORs. It should be tested statistically if the girls' OR differs from the boys', in order to draw the conclusion that the association is actually stronger for the girls. 

Response 8: We thank the reviewer for these comments. According to your suggestion, the interaction item between the child’s sex and PIU was first tested in a two-level multivariate logistic regression model, and the results demonstrated that the interaction item was significantly associated with sleep disturbance (P<0.001). This finding showed that although the 95% CI of the association between PIU and sleep disturbance for boys and girls overlapped, the difference between the association in girls and in boys was significant. These information have been added to the revised manuscript, and the reference of DeMaris has also been cited (please see page 3, lines 123-126; page 4, lines 145-148; page 6, lines 165-169; page 7, lines 194-196).

Point 9: Table 1: The percentages are presented differently in the columns. In the “Total” column, column percentages are used, while in the “Boys” and “Girls” columns, row percentages are used. The latter is confusing, and makes it difficult to compare the groups visually. I strongly suggest that column percentages are used overall.

Response 9: According to your suggestion, we have revised the Table 1 (please see the revised Table 1).

Discussion

Point 10: P5 General: Please summarize the main results (including the role of sex) early in the Discussion. Now, the results of one of the main research question (sex) is not presented until after some time (lines 180-181). However, here the correct terminology is used (“appeared stronger”) considering the lack of statistical testing of the potential difference of the ORs.

Response 10: We thank the reviewer for these Points. First, according to your suggestion, we have revised the first paragraph of the Discussion section (please see page 6, lines 175-183). Second, according to your previous Point (Point 8), the interaction item between the child’s sex and PIU was first tested in a two-level multivariate logistic regression model (please see the response of Point 8). Third, “appeared stronger” has been changed to “appeared slightly stronger” (please see page 7, line 213).

Point 11: P6, lines 181--. In relation to the gender differences, some biological mechanisms are discussed. However, there is no discussion about potential differences in the content of internet use or behavioral aspects of internet use. Do the girls and boys use the internet for different purposes or in different manner (e.g. long bouts vs many short uses, or daytime/nighttime etc)? I suggest that the discussion about potential mechanisms for gender differences is slightly broadened.

Response 11: According to your suggestion, we have revised the Discussion section to make it more clear (please see page 7, lines 216-220).

Point 12: General: The proportion of sleep disturbance (39%) in the study population is alarmingly high! Are sleep disturbances such a common problem for Chinese adolescents? Or is the cutoff at 7 for the PSQI perhaps not correct for a population of adolescents? However, I can see that if school starts earlier than 7:00 a.m., sleep deficiency is easily developed (especially since adolescents’ tend to have a circadium rhythm that favors late hours rather than early).

Response 12: Thank you for your questions. In the original manuscript, we have demonstrated that the Chinese version of PSQI was translated into Mandarin Chinese to better correspond to the meaning of the original item in PSQI by Chinese, and it has been shown to be valid and reliable, is also commonly used (please see page 3, lines 107-111). Additionally, the cutoff score has also been validated and widely used among Chinese adolescents, and our previous studies also demonstrated that the prevalence of sleep disturbance among Chinese adolescents was 39.6% [3].

Point 13: P6 line 186-- The authors propose several recommendations, based on the study results. Most of them are highly relevant. However, I do not see how number 5 is based on the study - (although relevant) it seems to relate more to general sleep behavior recommendations. 

Response 13: Thank you for your kind suggestion. We have deleted the fifth recommendation from this paragraph (please see page 7, line 229).

Conclusions

Point 14: See my previous notes about drawing conclusions about differences in ORs without statistically testing the difference. Either test the difference or tone down the conclusion.

Response 14: We thank the reviewer for these Points. According to your previous Point (Point 8), the interaction item between the child’s sex and PIU was first tested in a two-level multivariate logistic regression model (please see the response of Point 8). Additionally, we have revised the Conclusion section to make it more clear (please see page 8, lines 246-249).

References:

1.     Li H, Ji CY, Zong XN, Zhang YQ: Body mass index growth curves for Chinese children and adolescents aged 0 to 18  years (in Chinese). Zhonghua Er Ke Za Zhi. 2009, 47,493-498.

2.     Mei Z, Ogden CL, Flegal KM, Grummer-Strawn LM: Comparison of the prevalence of shortness, underweight, and overweight among US children aged 0 to 59 months by using the CDC 2000 and the WHO 2006 growth charts. J Pediatr. 2008, 153,622-628.

3.     Guo L, Deng J, He Y, Deng X, Huang J, Huang G, Gao X, Lu C: Prevalence and correlates of sleep disturbance and depressive symptoms among Chinese adolescents: a cross-sectional survey study. Bmj Open. 2014, 4,e5517.

Reviewer 3 Report

This study analyzes the relationship between PIU and gender disparities in sleep disturbances. A novel finding of the paper is that the association between PIU and sleep disturbance is found to be greater in girls than among boys.

There is increasing concern on the effects of sleep deprivation on health and academic performance. And there is growing evidence that exposure to blue-light and Internet use near bedtime may reduce sleep and sleep quality. In particular, there are growing concerns about the effect of problematic internet use (PIU) among teenagers. As previous study have already shown the relationship between problematic internet use and sleep disturbances among Chinese students (Chen et al., 2016), this study should further stress their relative contribution and focus more on the gender-differences.  Related to this, I have two main concerns:

1) The confidence intervals of the associations between PIU and sleep disturbance for boys and girls overlap (Table 1). Thus, the author should be more cautious in interpreting their findings

2) Women tend on sleep longer than men. If I am reading correctly Table 1 that’s true also in this sample. Thus the effect of PIU on sleep may be larger even if the relative effect with respect to the mean is similar. The authors should expand the discussion the summary statistics and  better link their study  to the existing literature on gender differnces in sleep deprivation.

Tables are not self-explanatory.

Relavant reference: A recent study has shown important effects of high-speed Internet on sleep, using an instrumental variable approach. This study should be quoted.

Billari, Francesco C., Osea Giuntella, and Luca Stella. "Broadband internet, digital temptations, and sleep." Journal of Economic Behavior & Organization 153 (2018): 58-76.

The work by Jean Twenge on the IGen should also be quoted.

Author Response

Response to Reviewer 3 Points

This study analyzes the relationship between PIU and gender disparities in sleep disturbances. A novel finding of the paper is that the association between PIU and sleep disturbance is found to be greater in girls than among boys. There is increasing concern on the effects of sleep deprivation on health and academic performance. And there is growing evidence that exposure to blue-light and Internet use near bedtime may reduce sleep and sleep quality. In particular, there are growing concerns about the effect of problematic internet use (PIU) among teenagers. As previous study have already shown the relationship between problematic internet use and sleep disturbances among Chinese students (Chen et al., 2016), this study should further stress their relative contribution and focus more on the gender-differences. Related to this, I have two main concerns:

Point 1: The confidence intervals of the associations between PIU and sleep disturbance for boys and girls overlap (Table 1). Thus, the author should be more cautious in interpreting their findings.

Response 1: Thank you for your carefully review of our manuscript. According to the suggestion of another reviewer, the interaction item between the child’s sex and PIU was first tested in a two-level multivariate logistic regression model, and the results demonstrated that the interaction item was significantly associated with sleep disturbance (P<0.001). This finding showed that although the 95% CI of the association between PIU and sleep disturbance for boys and girls overlapped, the difference between the association in girls and in boys was significant. These information has been added to the revised manuscript, and the reference of DeMaris has also been cited (please see page 3, lines 123-126; page 4, lines 145-148; page 6, lines 165-169; page 7, lines 194-196).

Point 2: Women tend on sleep longer than men. If I am reading correctly Table 1 that’s true also in this sample. Thus the effect of PIU on sleep may be larger even if the relative effect with respect to the mean is similar. The authors should expand the discussion the summary statistics and better link their study to the existing literature on gender differences in sleep deprivation.

Response 2: Thank you for your kind suggestion. We have added this information to the revised manuscript (please see page 7, lines 216-220).

Point 3: Tables are not self-explanatory.

Response 3: We thank the reviewer for this comment.

Point 4: Relavant reference: A recent study has shown important effects of high-speed Internet on sleep, using an instrumental variable approach. This study should be quoted. Billari, Francesco C., Osea Giuntella, and Luca Stella. "Broadband internet, digital temptations, and sleep." Journal of Economic Behavior & Organization 153 (2018): 58-76. The work by Jean Twenge on the IGen should also be quoted.

Response 4: According to your suggestion, we have cited the two references (please see page 2, line 46; page 6, line 185).

Round 2

Reviewer 2 Report

The authors have successfully considered all but one of my comments, and the manuscript has much improved! However, the authors seem to have misunderstood point 9 about Table 1. I will try to be clear. I apologize if I am over-explicit.

Table 1 is Sample characteristics stratified by sex. The table has 5 columns (column 2 = Total, column 3 = Boys, column 4 = Girls). The Total column presents column percentages of each categorical variable. But the Boys and Girls columns presents row percentages. This is counterintuitive – and makes it difficult to visually compare and understand the demographics/results without using a calculator. To show the characteristics stratified by sex, the column percentages for Boys and Girls need to be presented.

I will use Weight status as an example. In the Total column, normal = 84.7% and overweight/obesity = 15.3%. 84.7+15.3=100%. But in the Boys column, normal = 46% and overweight/obesity = 66.9%, 46+66.9=112.9%. In the Girls column normal = 54% and overweight = 33.1%. 54+33.1=87.1. If we want to understand the weight status stratified by sex, we need to see what percentage of the boys have normal weight and what percentage of the boys have overweight/obesity? And then the same for the girls. Using a calculator I find that 79% of the boys have normal weight and 21% are overweight (=100). And for the girls: 90 % normal weight and 10% overweight (=100). These are the percentages that should be presented. The row percentages make no sense.

So, my point is that I strongly suggest that Table 1 is revised so that column percentages are presented for all categorical variables in the Boys and Girls columns (instead of row percentages). Note that the first row (presenting the n of the columns) can keep the previous manuscript version (of row percentages), i.e., that boys represent 49,2 % of the sample and girls represent 50,8 %. 

In addition, note that there seems to be 6 more boys and 6 more girls in, for example, Weight status, than in the total n of the sample. 

Author Response

Response to Reviewer 2 Points

Point 1: The authors have successfully considered all but one of my comments, and the manuscript has much improved! However, the authors seem to have misunderstood point 9 about Table 1. I will try to be clear. I apologize if I am over-explicit.

Table 1 is Sample characteristics stratified by sex. The table has 5 columns (column 2 = Total, column 3 = Boys, column 4 = Girls). The Total column presents column percentages of each categorical variable. But the Boys and Girls columns presents row percentages. This is counterintuitive – and makes it difficult to visually compare and understand the demographics/results without using a calculator. To show the characteristics stratified by sex, the column percentages for Boys and Girls need to be presented.

I will use Weight status as an example. In the Total column, normal = 84.7% and overweight/obesity = 15.3%. 84.7+15.3=100%. But in the Boys column, normal = 46% and overweight/obesity = 66.9%, 46+66.9=112.9%. In the Girls column normal = 54% and overweight = 33.1%. 54+33.1=87.1. If we want to understand the weight status stratified by sex, we need to see what percentage of the boys have normal weight and what percentage of the boys have overweight/obesity? And then the same for the girls. Using a calculator I find that 79% of the boys have normal weight and 21% are overweight (=100). And for the girls: 90 % normal weight and 10% overweight (=100). These are the percentages that should be presented. The row percentages make no sense.

So, my point is that I strongly suggest that Table 1 is revised so that column percentages are presented for all categorical variables in the Boys and Girls columns (instead of row percentages). Note that the first row (presenting the n of the columns) can keep the previous manuscript version (of row percentages), i.e., that boys represent 49,2 % of the sample and girls represent 50,8 %. In addition, note that there seems to be 6 more boys and 6 more girls in, for example, Weight status, than in the total n of the sample.

Response 1: We truly appreciate your time in reviewing our manuscript, and thank you for your patient explanation. According to you suggestion, we have revised the Table 1 (please see the revised Table 1). 

Reviewer 3 Report

The authors addressed most of my comments.

Author Response

Response to Reviewer 3 Points

Point 1: The authors addressed most of my comments.

Response 1: We would like to thank you for the time and effort taken to review our manuscript.
